# Cortical entrainment to hierarchical contextual rhythms recomposes dynamic attending in visual perception

Peijun Yuan[1,2,3], Ruichen Hu[1,2,3], Xue Zhang[1,2,3], Ying Wang[1,2,3]*, Yi Jiang[1,2,3,4]*

[1]State Key Laboratory of Brain and Cognitive Science, CAS Center for Excellence in Brain Science and Intelligence Technology, Institute of Psychology, Chinese Academy of Sciences, Beijing, China; [2]Department of Psychology, University of Chinese Academy of Sciences, Beijing, China; [3]Chinese Institute for Brain Research, Beijing, China; [4]Institute of Artificial Intelligence, Hefei Comprehensive National Science Center, Hefei, China

**Abstract** Temporal regularity is ubiquitous and essential to guiding attention and coordinating behavior within a dynamic environment. Previous researchers have modeled attention as an internal rhythm that may entrain to first-order regularity from rhythmic events to prioritize information selection at specific time points. Using the attentional blink paradigm, here we show that higher-order regularity based on rhythmic organization of contextual features (pitch, color, or motion) may serve as a temporal frame to recompose the dynamic profile of visual temporal attention. Critically, such attentional reframing effect is well predicted by cortical entrainment to the higher-order contextual structure at the delta band as well as its coupling with the stimulus-driven alpha power. These results suggest that the human brain involuntarily exploits multiscale regularities in rhythmic contexts to recompose dynamic attending in visual perception, and highlight neural entrainment as a central mechanism for optimizing our conscious experience of the world in the time dimension.

*For correspondence:
wangying@psych.ac.cn (YW);
yijiang@psych.ac.cn (YJ)

**Competing interests:** The authors declare that no competing interests exist.

## Introduction

Deploying attention over time is crucial for guiding human activities within a rapidly changing environment. However, the constant influx of information goes far beyond our mental capacity, impeding even the most competent human brain from capturing every nuance of the details. How does the human brain surmount such limitations in temporal attention allocation during dynamic information processing?

One feasible solution, as that for spatial attention, is through selection, or by shining an attentional 'spotlight' on the most relevant information while filtering out the irrelevant regarding the task demands (*Posner, 1980*). When it comes to the temporal domain, people tend to utilize regularities in the sensory information flow for directing attention to the moments when a target event is expected to occur (*Nobre et al., 2007*; *Nobre and van Ede, 2018*). As a great example, Jones and colleagues have shown in a series of studies that after listening to a rhythmic tone sequence, auditory perception in terms of pitch judgment and time discrimination was more accurate for target tones appearing at the expected than the unexpected time points (*Jones et al., 2002*; *Large and Jones, 1999*). Such facilitation effects have been extended to various aspects of visual perception and even across sensory modalities (*Bolger et al., 2014*; *Brochard et al., 2013*; *Mathewson et al., 2010*; *Miller et al., 2013*; *ten Oever et al., 2014*), implicating the involvement of a general attentional selection mechanism guided by the regularity in stimulus timing.

In this line of studies, perceptual responses were significantly improved for targets appearing within a rhythmic context but not within an arrhythmic context. These findings can be interpreted by

the dynamic attending theory (DAT), which assumes attention as an internal oscillatory activity (or attending rhythm) that can be entrained to rhythmic structures of the exogenous events (*Jones, 1976*; *Jones et al., 1981*; *Jones and Boltz, 1989*; *Large and Jones, 1999*). In line with this assumption, electrophysiological research in humans and non-human primates have found entrainment of intrinsic neural oscillations to external stimulus rhythms, and regarded such process as an instrument for selective attention (*Calderone et al., 2014*; *Obleser and Kayser, 2019*; *Schroeder and Lakatos, 2009*). Through neural entrainment, neuronal excitability aligns with the occurrence of rhythmic events, creating 'temporal attentional spotlights' that attract the brain's attentional resources toward a string of selected moments (*Calderone et al., 2014*; *Henry and Herrmann, 2014*; *Lakatos et al., 2008*; *Lakatos et al., 2013*; *Schroeder and Lakatos, 2009*).

The synchronization between the internal attending rhythm and the external rhythms allows us to direct attention proactively and enhance perception at the anticipated moments. With regard to forming a coherent perception of the dynamic environment, however, we should not only select information bound to the anticipated time points, but also allocate attentional resources among these points, raising the problem of dynamic attentional deployment over an information stream. For instance, when viewing a rapid serial visual presentation (RSVP) stream, there is a large chance that the observer would miss the second of two temporally proximate targets, as the allocation of attention to the first target hinders the redeployment of mental resources to the second one (*Broadbent and Broadbent, 1987*). This phenomenon, vividly referred to as the attentional blink (AB) (*Chun and Potter, 1995*; *Raymond et al., 1992*), has attracted much interest as it reveals the limitations of attentional allocation and memory processes that may become a bottleneck for conscious awareness (*Dux and Marois, 2009*; *Martens and Wyble, 2010*; *Shapiro et al., 1997*). More intriguingly, as items in the RSVP stream are all rhythmically presented and temporally predictable, the AB effect poses a challenge in dynamic attending that cannot be circumvented solely by the anticipation built upon stimulus timing.

To address this challenge, here we propose that, the brain has to rely, as a complement to the first-order regularity in rhythmic stimulation, on regularities in the higher-order temporal structure of the information stream. More specifically, if the endogenous attentional rhythm could entrain automatically not only to the stimulus rhythm but also to the higher-order structure based on the information content, the deployment of temporal attention might be reconstructed in a way that facilitates target detection in the AB task. To test this hypothesis, we synchronized the original AB stream (stimulation rate at 10 Hz) to a hierarchical contextual stream that possessed a feature-based temporal structure—a 2.5 Hz rhythm arising from periodic changes of a physical feature, superimposed on its stimulus rhythm at 10 Hz. Using temporal structures defined by a variety of features (pitch, color, etc.), we provided converging evidence that the structured context, which was task-irrelevant and even from a different modality, could regulate the dynamic deployment of visual attention so as to alleviate the AB effect. To further unravel the neural basis of the observed attentional modulation effect, we conducted an electroencephalogram (EEG) experiment. We are particularly interested in whether neural oscillations can entrain to the contextual temporal structure of stimulus feature along with that of stimulus onset timing, and more critically, whether and how the cortical entrainment to these hierarchical structures mediates the behavioral modulation effect.

## Results

### Temporal structure of contextual auditory stream recomposes visual attentional deployment

In Experiment 1a, we first explored whether feature-defined temporal structure from a contextual auditory stream could regulate visual attentional deployment during the AB task. If so, the AB effect should be modulated by the positions of the visual targets relative to the structure-defined cycles arising from periodic changes of the background sound (*Figure 1A*, see Materials and methods for details). Above all, we found a robust AB effect for Experiment 1a, as well as for the other experiments reported in the current study. The accuracy of T1 performance was very high in both the baseline session (mean ±SE: 0.959 ± 0.008) and the context session (0.953 ± 0.006), and the overall T1 performance ranged from 0.907 to 0.972 in all experiments. By contrast, the T2 detection accuracy conditioned on correct T1 response was generally impaired in the short-SOA conditions relative to

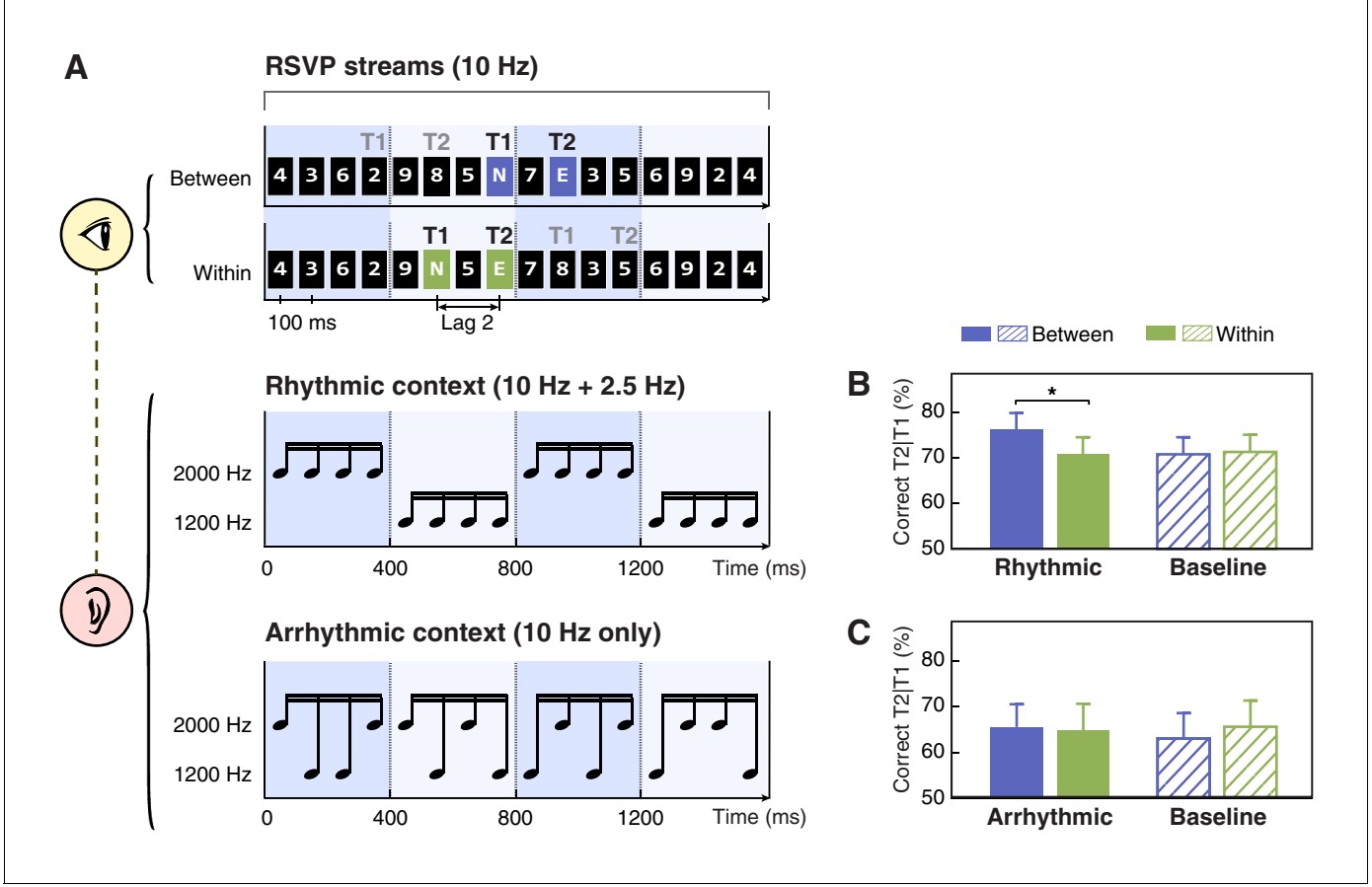

**Figure 1.** Schematics of stimuli and results for Experiments 1a and 1b. (**A**) In the AB task, participants were presented with rapid serial visual presentation (RSVP) streams at 10 Hz (top). Each stream contained two capital letter targets embedded in fourteen number distractors. Black and gray 'T1' and 'T2' denote two alternative options for target locations in the short-SOA conditions. These targets were located either in two adjacent cycles (the between-cycle condition, displayed on violet background for illustration only) or within the same rhythmic cycle (the within-cycle condition, displayed on green background for illustration only) defined by a rhythmic auditory context in Experiment 1a (middle). Arrhythmic context was used as a control in Experiment 1b (bottom). (**B and C**) T2 detection accuracy conditioned on correct T1 response for the experiments using rhythmic and arrhythmic contexts. Note that in the baseline (visual-only) session, the labels of 'between' and 'within' were used to refer to the conditions where the two targets shared the same absolute positions with their corresponding conditions in the context (audiovisual) session. Error bars represent 1 SEM; *p<0.05.

The online version of this article includes the following source data for figure 1:

**Source data 1.** T2 detection accuracy for individual participants in Experiments 1a and 1b.

that in the long-SOA condition (mean accuracy >0.9 for all experiments), during both the context session ($t(15) = 5.443$, p<0.001, Cohen's $d = 1.361$) and the baseline session ($t(15) = 6.720$, p<0.001, Cohen's $d = 1.680$).

More importantly, looking close at T2 performance in the short-SOA conditions (*Figure 1B*), we found T2 was better identified when two targets appeared in two adjacent cycles (between-cycle condition) than within the same cycle defined by the background sounds (within-cycle condition). Notably, such difference was observed only for the context session ($t(15) = 2.947$, p=0.010, Cohen's $d = 0.737$) but not for the baseline (no sound) session ($t(15) = -0.212$, p=0.835, Cohen's $d = 0.053$), although the target positions were completely matched between these two sessions. Meanwhile, only in the between-cycle condition, the contextual sounds enhanced T2 detection accuracy relative to the baseline ($t(15) = 2.287$, p=0.037, Cohen's $d = 0.572$), while in the within-cycle condition, the performance kept comparable between the context and baseline sessions ($t(15) = -0.271$, p=0.790, Cohen's $d = 0.068$). The observed dissociation was further confirmed by a two-way repeated-measures ANOVA, which yielded a significant interaction between experimental session (baseline vs.

context session) and target position (between- vs. within-cycle, defined by the context) ($F$(1, 15)=7.151, p=0.017, $\eta_p^2$ = 0.323).

Results from Experiment 1a demonstrated that feature-based temporal structure of an auditory stream, although being task-irrelevant, could systematically modulate the allocation of visual attention over the AB stream. Since the temporal structure of the contextual sounds was defined by periodic changes of pitch, when two targets were located in distinct cycles as in the between-cycle condition, they were accompanied by different tones, in contrast to that when located within the same cycle they were accompanied by the same tone. It is possible that the contrast of physical stimulation (i.e. pitch) at T1 and T2 could account for the performance improvement in the between-cycle condition. To test this possibility, in Experiment 1b, we matched the pitch of tones at target occurrence with that in Experiment 1a for the between- and within-cycle condition respectively, whereas disrupted feature-based regularity in the temporal structure of the contextual sound sequence (*Figure 1A*, bottom). Despite that the sounds paired with the targets were exactly the same as in Experiment 1a, the difference in T2 detection accuracy caused by the contextual sounds was no longer observed ($t$(15) = 0.433, p=0.671, Cohen's $d$ = 0.108), neither was its interaction with experimental session ($F$(1, 15)=2.734, p=0.119, $\eta_p^2$ = 0.154; *Figure 1C*). In other words, T2 was identified with similar accuracy across all the conditions in Experiment 1b, suggesting that it is the temporal structure of the contextual sounds, not the pitch difference at target presentation, that accounts for the between-cycle facilitation effect observed in Experiment 1a.

## Generalization of the modulation effect to different cycle frequencies

In Experiment 1a, the auditory context always changed its pitch value every four items, that is every 400 ms as one cycle, resulting in rhythmic cycles at 2.5 Hz. In Experiment 1 c, we tested whether the modulation effect we observed could be generalized to other cycle frequencies. We set the pitch change rate to 2 Hz (i.e. five items per cycle; *Figure 2A*, upper) and 3.3 Hz (i.e. three items per cycle; *Figure 2A*, lower). For both context frequencies, the T2 detection performance in the between-cycle condition was significantly higher than that in the within-cycle condition (*Figure 2B*; for 2 Hz, $t$(15) = 3.478, p=0.004, Cohen's $d$ = 0.869; for 3.3 Hz, $t$(15) = 2.467, p=0.030, Cohen's $d$ = 0.617), suggesting successful attentional modulation effects. Furthermore, a repeated-measures ANOVA on T2 accuracy revealed only a significant main effect of relative target position (i.e. between- vs. within-cycle) ($F$(1, 15)=23.320, p<0.001, $\eta_p^2$ = 0.609), with a marginally significant main effect of frequency ($F$(1, 15)=4.337, p=0.055, $\eta_p^2$ = 0.224) and no interaction between these two factors ($F$(1, 15)=0.204, p=0.658, $\eta_p^2$ = 0.013).

## The effect of temporal attention rather than perceptual grouping

As temporal structure of the context was constructed by auditory items sharing the same feature (i.e. pitch), one may argue that perceptual grouping on the basis of similarity (*Bregman, 1994*), instead of dynamic attending guided by feature-based temporal regularities, contributes to the between-cycle benefit that we observed. To disentangle these factors, in Experiment 1d, we changed the pitch value of tone sequences irregularly to form auditory streams that could be grouped in varying lengths (*Figure 2C*, upper). Although temporal grouping was reserved in this setting, no facilitation effect was observed when targets were separated in two distinct groups relative to when they were displayed within the same group (*Figure 2D*, Irreg-G). T2 detection performance was comparable in the between- and the within-group conditions ($t$(15) = 0.348, p=0.733, Cohen's $d$ = 0.087).

Compared with Experiments 1a and 1 c, the strength of temporal grouping in Experiment 1d might be attenuated due to irregular number of items in each group, which could lead to the lack of behavioral modulation effect. To solve this issue, in Experiment 1e (*Figure 2C*, lower), we changed the pitch every four items to keep the rule of temporal grouping exactly the same as that in Experiment 1a. Nevertheless, we disrupted the regularity of stimulus timing. Such manipulation would have a detrimental impact on dynamic deployment of temporal attention in general, according to the basic assumption of the DAT (*Jones et al., 1982*; *Jones and Boltz, 1989*; *Large and Jones, 1999*). On the other hand, it would have little influence on the grouping effect. Therefore, if temporal attention rather than perceptual grouping is essential to the behavioral modulation effect observed in the current study, we should expect such effect to disappear in Experiment 1e. In line

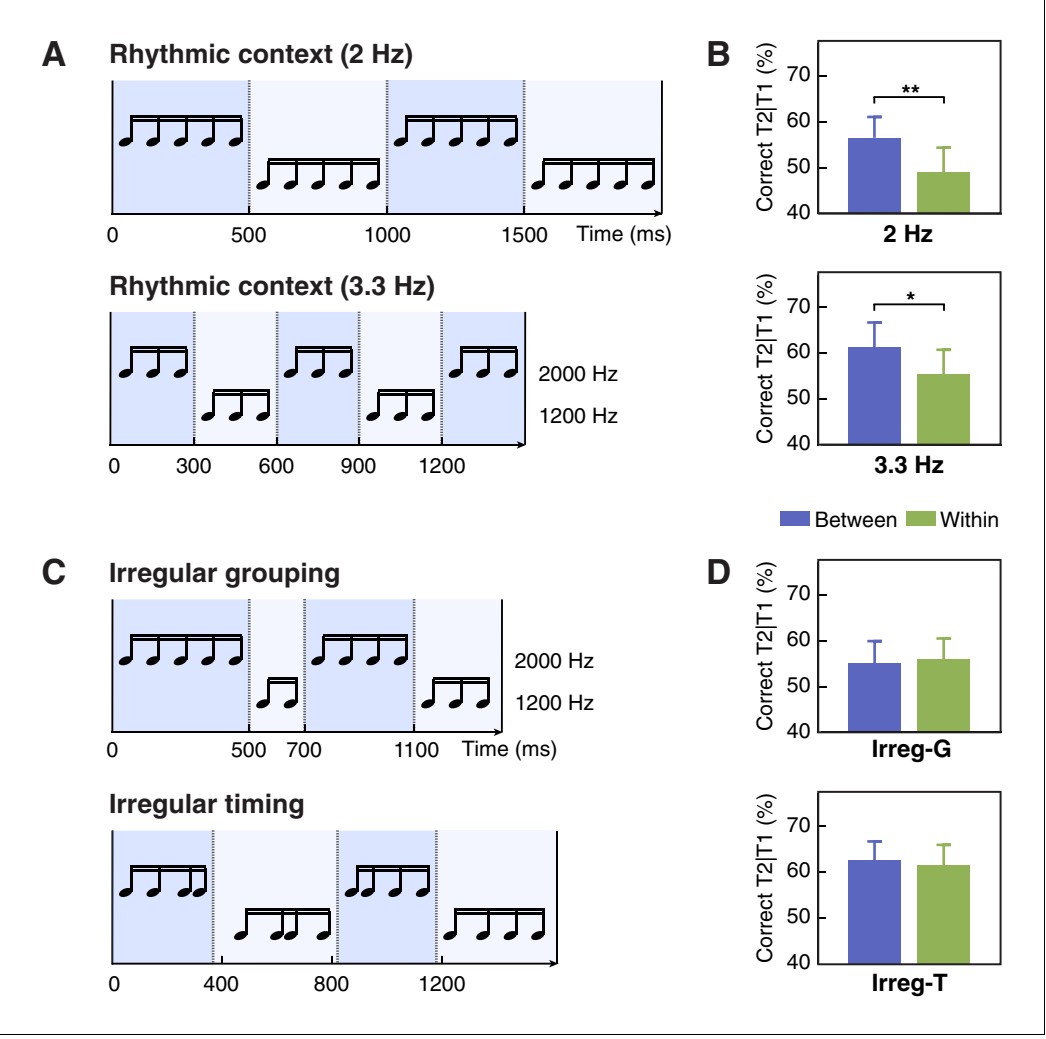

**Figure 2.** Stimuli and results for Experiments 1 c, 1d and 1e. (**A**) Contextual tone sequences with pitch changed every five tones (2 Hz, upper) and every three tones (3.3 Hz, lower) in Experiment 1 c. (**B**) T2 performance in short-SOA conditions for 2 Hz(upper) and 3.3 Hz (lower) sequence in Experiment 1 c. (**C**) The auditory context was grouped irregularly into four chunks with different numbers of tones (Irreg-G) in Experiment 1d (upper) and into four regular chunks (four tones in each) but with irregular onset timing (Irreg-T) in Experiment 1e (lower). (**D**) T2 performance in Experiment 1d (upper) and 1e (lower). Error bars represent 1 SEM; *p<0.05, **p<0.01.

The online version of this article includes the following source data for figure 2:

**Source data 1.** T2 detection accuracy for individual participants in Experiments 1 c, 1d, and 1e.

with our speculation, when the stimulus onset timing was randomized, T2 detection performance in the between-cycle condition was no longer improved relative to the within-cycle condition (*Figure 2D*, Irreg-T; $t(15) = 0.302$, p=0.767, Cohen's $d = 0.076$), despite the potential benefit of the grouping effect. Putting together, the absence of context-induced modulation effect in Experiments 1d and 1e consistently supports the idea that temporal grouping without dynamic attending guided by feature- and timing-related regularities in the auditory context is insufficient to cause the behavioral modulation effect.

## Temporal regularities in color-defined rhythmic structure recompose visual attentional deployment

Information from the auditory modality, like speech and music, is inherently organized in time and provides rich sources of rhythmic structures that can be proactively tracked by the human brain (*Arnal and Giraud, 2012*; *Doelling and Poeppel, 2015*; *Haegens and Zion Golumbic, 2018*;

*Zion Golumbic et al., 2012*). This suggests a possibility that the role of rhythmic structure in guiding attention is exclusive to auditory context, which may explain the findings from Experiment 1 that temporal structures generated by rhythmic changes of auditory signals in the background automatically modulate the AB effect. To test this idea, we designed Experiment 2 to directly investigate whether temporal structures based on the change of visual properties would exert a similar influence on temporal attentional deployment. In Experiment 2a, we used visual patterns with periodic change in background color as the temporal context while observers were performing the same AB task (*Figure 3A*). As a control experiment, Experiment 2b followed the same logic for Experiment 1b, in which we destroyed the structure of the visual context by changing the background color in random orders, but kept the background color presented with the targets the same as that in Experiment 2a (*Figure 3C*).

Similar to findings obtained from Experiment 1a, the interaction between experimental session (baseline vs. context session) and target position (between- vs. within-cycle) was significant in Experiment 2a (*Figure 3B*; $F(1, 15)=5.180$, $p=0.038$, $\eta_p^2 = 0.257$). In the context session only, T2 performance in the between-cycle condition was better than that in the within-cycle condition ($t(15) = 3.538$, $p=0.003$, Cohen's $d = 0.885$). Compared with the baseline session, T2 performance was only improved in the between-cycle condition ($t(15) = 2.274$, $p=0.038$, Cohen's $d = 0.569$). By contrast, in Experiment 2b, we did not observe a significant facilitation effect in the between-cycle condition compared with the within-cycle condition ($t(15) = -1.176$, $p=0.258$, Cohen's $d = 0.294$) or with its counterpart in the baseline session ($t(15) = 0.685$, $p=0.504$, Cohen's $d = 0.171$), nor did we observe the interaction between experimental session and target position (*Figure 3D*; $F(1, 15)=1.435$, $p=0.250$, $\eta_p^2 = 0.087$). These findings suggest that the utilization of feature-based temporal

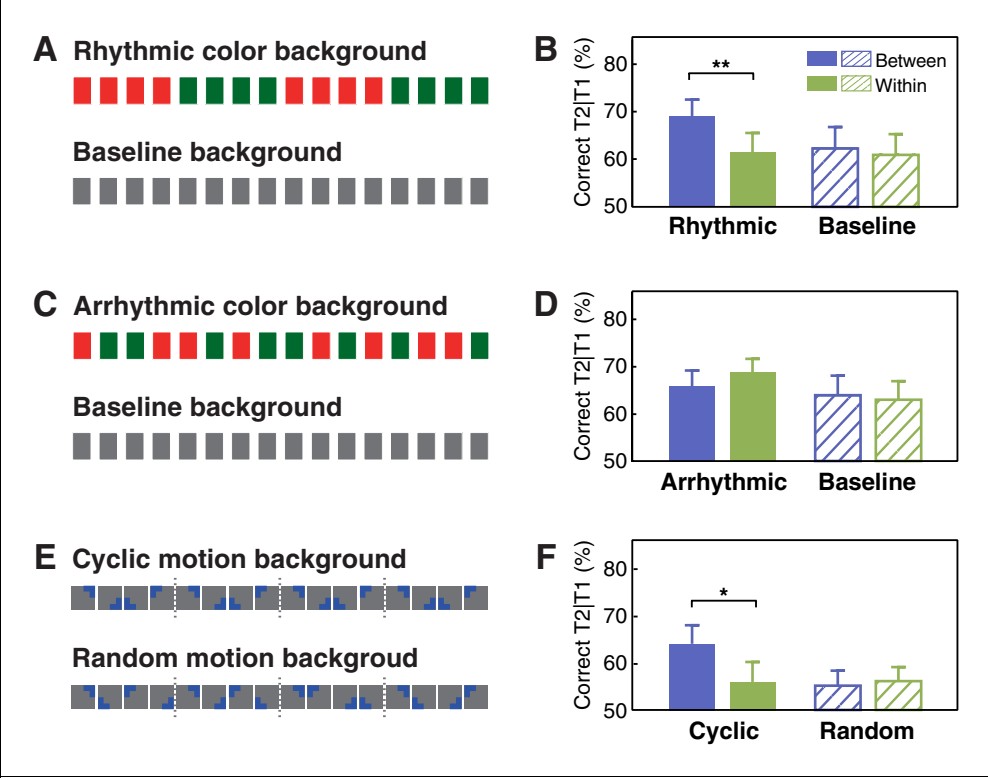

**Figure 3.** Stimuli and results for Experiments 2 and 3 using the visual contexts. (A) The visual context with or without periodic changes in the background color and (B) the T2 performance in Experiment 2a. (C) The visual context with or without the background color changed irregularly and (D) the T2 performance in Experiment 2b. (E) Contextual rhythms defined by cyclic/random motion at a constant speed and (F) the T2 performance in Experiment 3. Error bars represent 1 SEM; *p<0.05, **p<0.01.

The online version of this article includes the following source data for figure 3:

**Source data 1.** T2 detection accuracy for individual participants in Experiments 2a, 2b, and 3.

regularities in attentional guidance is a fundamental principle that holds true not only for auditory but also for visual processing.

## Excluding the impact of structure boundary: evidence from motion context

So far, results from Experiments 1 and 2 have demonstrated a general regulatory effect that feature-based temporal structure from task-irrelevant information recomposed visual attentional allocation during the AB task, which could be exerted within the same or cross different sensory modalities. In both experiments, however, the switch from one feature-based rhythmic cycle to another was always accompanied by an abrupt change in physical features (pitch or color), resulting in an explicit boundary before T2 presentation in the between-cycle but not in the within-cycle condition. This abrupt change may serve as an attentional cue or alerting signal for the upcoming T2, and thus accounts for the improvement of performance in the between-cycle condition. To examine this possibility, in Experiment 3, we introduced a cyclic motion context that possessed feature-based rhythmicity identical to those contextual rhythms in previous experiments (for more details, see Materials and methods) but had no abrupt boundaries between cycles (*Figure 3E*). Once again, we observed significant improvement of T2 performance in the between-cycle condition relative to the within-cycle condition in the cyclic motion session (*Figure 3F*; $t(15) = 2.674$, p=0.017, Cohen's $d = 0.669$), but this was not the case in the random motion session ($t(15) = -0.330$, p=0.746, Cohen's $d = 0.082$), resulting in a significant interaction between experimental session (random vs. cyclic motion session) and target position (between- vs. within-cycle): $F(1, 15)=9.253$, p=0.008, $\eta_p^2 = 0.382$. These results provide compelling evidence that explicit perceptual boundaries are not necessary for the temporal structure in the context to regulate the allocation of attentional resources.

## EEG experiment: the role of neural entrainment in regulating attentional deployment

### Neural tracking of higher-order temporal structure of contextual rhythms predicts the behavioral modulation effect

To investigate the neural mechanisms underlying the observed context-induced effect, we carried out an EEG experiment using the same task as that in Experiment 1a. First of all, we replicated the behavioral modulation effect that T2 performance was significantly better in the between-cycle condition versus the within-cycle condition, only in the context session (between-cycle: $0.567 \pm 0.036$, within-cycle: $0.520 \pm 0.039$, $t(15) = 3.838$, p=0.002, Cohen's $d = 0.960$) but not in the baseline session (between-cycle: $0.519 \pm 0.039$, within-cycle: $0.527 \pm 0.043$, $t(15) = 0.296$, p=0.771, Cohen's $d = 0.074$). Furthermore, to identify the oscillatory characteristics of EEG signals in response to stimulus rhythms, we examined the FFT spectral peaks by subtracting the mean power of two nearest neighboring frequencies from the power at the stimulus frequency. Power spectrum in *Figure 4A* shows several peaks for the context session, with the highest at 10 Hz (compared with zero using one-sample *t*-test, right-tailed, $t(15) = 10.610$, p<0.001, FDR-corrected for multiple comparisons across frequencies) corresponding to the common stimulation frequency of the visual and auditory streams. More importantly, the second-highest peak appeared at 2.5 Hz ($t(15) = 5.730$, p<0.001, FDR-corrected), followed by its harmonics at 5 and 7.5 Hz, indicating neural tracking of the feature-defined structure of the auditory context. In contrast with the observation in the context session, we only found significant power peak at 10 Hz ($t(15) = 9.405$, p<0.001, FDR-corrected), but not at 2.5 Hz ($t(15) = 0.301$, p=0.384, FDR-corrected) in the baseline session where contextual rhythms were absent, and the power at 2.5 Hz was significantly weaker than that in the context session ($t(15) = 3.421$, p=0.002, FDR-corrected).

The significant enhancement of EEG power at 2.5 Hz clearly demonstrates that the brain can entrain to the higher-order structure defined by changes in an auditory feature (i.e. pitch) of the contextual stream. Consistent with previous studies, we also observed a wide range of individual variation in such cortical tracking of contextual rhythms (*Grahn and McAuley, 2009*; *Kranczioch, 2017*; *Nozaradan et al., 2016*). Could such variation predict one's ability to extract and utilize the feature-based structure at the neural level, and thus explain the individual differences in the attentional modulation effect? To explore this possibility, we calculated the Pearson correlation between the magnitude of the neural entrainment effect and the behavioral modulation index (BMI) using a cluster-

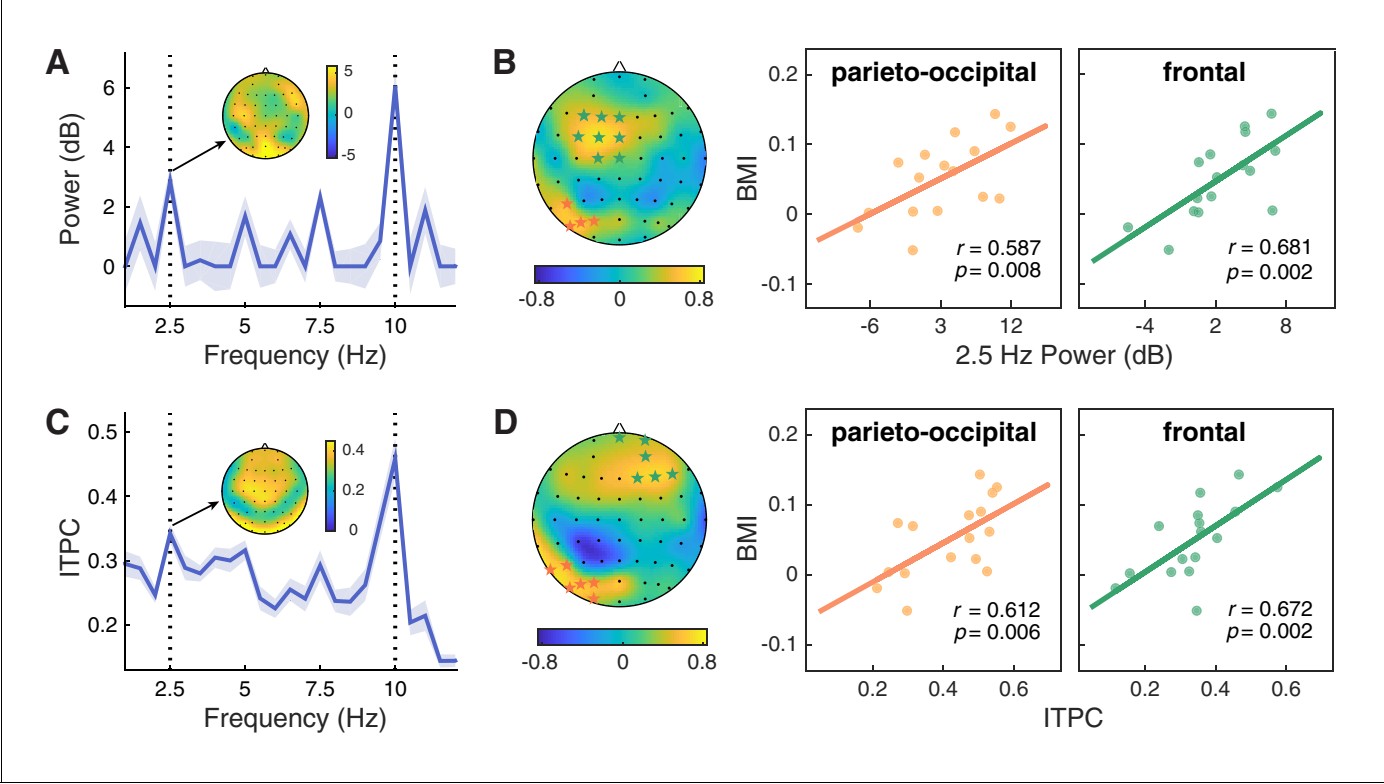

**Figure 4.** Neural entrainment to contextual rhythms and its correlation with the attentional modulation effect. (**A**) The power spectrum of EEG signals averaged across all epochs and channels. For each frequency, power was normalized by subtracting the mean power of the two nearest neighboring frequencies from the power of the center frequency. Shaded areas indicate standard errors of the mean. (**B**) The 2.5 Hz power entrainment effect at the parieto-occipital cluster and the frontal cluster, as respectively indicated by orange and green stars in the scalp topographic map, significantly correlated with the behavioral modulation index (BMI). (**C** and **D**) Analysis of inter-trial phase coherence (ITPC) results yielded similar patterns to those for power.

The online version of this article includes the following source data and figure supplement(s) for figure 4:

**Source data 1.** Source data for *Figure 4* and *Figure 4—figure supplement 1*.

**Figure supplement 1.** Neural entrainment to contextual rhythms indexed by induced power and its correlation with the attentional modulation effect.

based permutation test. In the context session, we identified two significant clusters showing positive correlation between power at 2.5 Hz and individuals' behavioral effect—one in the parieto-occipital region (*Figure 4B*; P5, PO7, PO5, PO3; $r = 0.587$, p=0.008, right-tailed) and the other in the frontal area (F3, F1, FZ, FC3, FC1, FCZ, C1, CZ; $r = 0.681$, p=0.002). By contrast, no significant clusters were found in the baseline session (p>0.05).

To further examine the role of brain activity phase-locked with the rhythmic context, we also analyzed the inter-trial phase coherence (ITPC) of EEG signals. Consistent with the power spectrum, ITPC in the context session peaked at 2.5 and 10 Hz (*Figure 4C*), suggesting a hierarchical entrainment effect elicited by both feature-based and time-based regularities. By contrast, ITPC in the baseline session only peaked at 10 Hz, mirroring the stimulation rate of the visual stream, and the ITPC at 2.5 Hz was significantly weaker than that in the context session ($t(15) = 4.652$, p<0.001, FDR-corrected). Critically, only in the context session, the 2.5 Hz ITPC was positively correlated with the behavioral modulation index, yielding two significant clusters in the parieto-occipital area (*Figure 4D*; P7, P5, PO7, PO5, PO3, O1: $r = 0.612$, p=0.006) and the frontal area (FPZ, FP2, AF4, F2, F4, F6; $r = 0.672$, p=0.002). In addition to the above analysis of phase-locked neural responses, we also looked into the power spectrum based on the average of single-trial spectral transforms, that is the induced power, which puts emphasis on the intrinsic non-phase-locked activities. In line with the results of evoked power and ITPC, we found consistent patterns for the induced power (for details see *Figure 4—figure supplement 1*). Taken together, the results of 2.5 Hz power and ITPC

jointly demonstrate that the better one's brain oscillations entrain to the higher-order temporal structure of the contextual rhythms, the larger attentional enhancement one may exhibit in the between-cycle condition over the within-cycle condition.

## Neural responses to first-order rhythms at 10 Hz reflect the attentional modulation

Alpha oscillations have been considered to play a crucial and even causal role in temporal attention, particularly in the AB effect (*Hanslmayr et al., 2011*; *Klimesch, 2012*). As the AB phenomenon is characteristic of its stimulation frequency approximately at 10 Hz within the alpha band, the brain can be in a resonant state with the AB stream at the same frequency. It has been demonstrated that an increase in alpha power at the stimulus frequency indicated attentional orienting to the stimulus stream, providing an online measure of attentional allocation over the RSVP stream (*Müller and Hübner, 2002*). On the other hand, enhanced alpha power at stimulus rate in the AB task has also been shown to be associated with correct T2 detection (*Janson et al., 2014*; *Keil et al., 2006*). Motivated by these findings, we investigated whether the 10 Hz alpha activity related to T2 processing could reflect the attentional modulation in our study. We calculated alpha power around the stimulation frequency (9.5–10.5 Hz) within the time window of 0–100 ms after T2 onset, and found two significant clusters for the context session—one in the left parieto-occipital region (*Figure 5A*; T7, C5, C3, TP7, CP5, CP3, P5, P3, PO5, PO3, O1) and the other in a right-lateralized region (AF4, F2, F4, FC4, FC6, FT8, C4, C6, T8, CP4, CP6, TP8, P8), both showing stronger 10 Hz alpha power in the between-cycle condition than in the within-cycle condition (for the left cluster, $t(15) = 3.570$, p=0.0014; for the right cluster, $t(15) = 3.631$, p=0.0012, right-tailed, cluster-based permutation test). To verify that the observed modulation effect was due to context-induced entrainment rather than a by-product of post-T2 processing, we further examined the 10 Hz alpha power within the time window of $-100$–0 ms prior to T2 onset. Results revealed an enhancement of this pre-T2 neural response for the between-cycle condition relative to the within-cycle condition, which is similar to that observed within the post-T2 time window but more restricted to the left parieto-occipital cluster (*Figure 5A*; CP3, CP5, P3, P5, PO3, PO5, POZ, O1, OZ; $t(15) = 2.774$, p=0.007).

## Cross-frequency coupling between delta phase and alpha power correlates with the attentional modulation effect

Examinations on delta-band entrainment effect, and 10 Hz alpha power both reveal behavioral relevance in our study. This leads to a natural question of whether the observed attentional modulation effect is implemented through a coordinative process between neural oscillations at delta and alpha bands. To address this question, we analyzed cross-frequency coupling between delta phase and alpha power, which has been found to support the attentional selection between competing stimuli (*Gomez-Ramirez et al., 2011*; *Wilson and Foxe, 2020*; *Wöstmann et al., 2016*). We conducted the analysis in two clusters whose neural responses in both the delta band (the ITPC at 2.5 Hz) and the alpha band (10 Hz alpha power) had an established link with the attentional modulation effect: one in the parieto-occipital region (P5, PO3, PO5, O1) and the other in the frontal region (AF4, F2, F4). We calculated the modulation index (MI) of phase-amplitude coupling (PAC) between delta (1.5–3.5 Hz) and alpha band (7–13 Hz) for each cluster. The MI was stronger in the between-cycle condition than in the within-cycle condition, while the effect reached significance only in the parieto-occipital region (*Figure 5B*; $t(15) = 2.432$, p=0.028) but not in the frontal region ($t(15) = 1.459$, p=0.165). More importantly, this contrast effect of delta-alpha PAC showed a positive correlation with the attentional modulation effect on behavioral performance, which was also restricted to the parieto-occipital region (*Figure 5C*; $r = 0.660$, p=0.005) and not found in the frontal region ($r = 0.154$, p=0.569). To further confirm the association between the delta-alpha PAC and the observed attentional modulation effect, we did a cluster-based permutation test, which again yielded a positively significant cluster in the parieto-occipital region (PO7, PO5, PO3, O1, OZ; $r = 0.697$, p=0.003). These results, combined with the findings from single-band analyses, indicate that cortical tracking of hierarchical temporal structures of the auditory context, as well as the coordination of such cortical tracking effects in delta and alpha bands, may play a vital role in reconstructing the deployment of visual attention in the AB task.

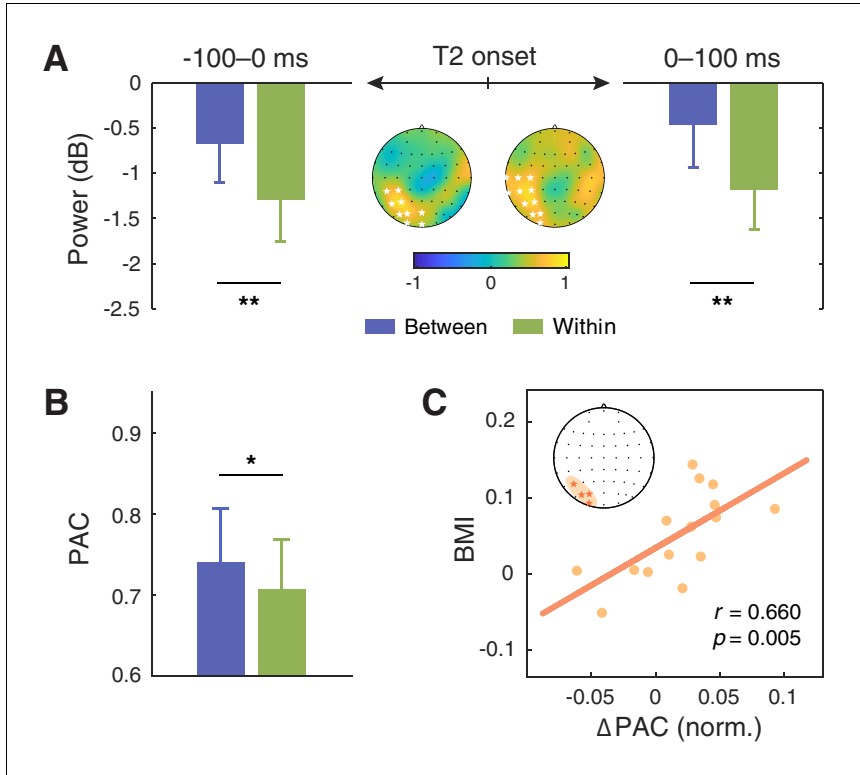

**Figure 5.** Modulation effect of the alpha power and its coupling with the delta phase. (**A**) 10 Hz alpha power averaged within the time window of −100–0 ms (left) and 0–100 ms (right) relative to the T2 onset was significantly higher in the between-cycle condition than in the within-cycle condition in a left parieto-occipital cluster (indicated by white stars). (**B**) The modulation index of phase-amplitude coupling (PAC) between the delta and alpha bands was higher for the between-cycle condition than for the within-cycle condition, and (**C**) the difference in normalized PAC strength could predict the BMI across individuals. Shadowed area in the topographic plot indicates the cluster showing significant behavioral relevance in both delta- and alpha-band activities. Error bars represent 1 SEM; *p<0.05, **p<0.01.

The online version of this article includes the following source data for figure 5:

**Source data 1.** Source data for *Figure 5*.

## Discussion

### Temporal attention guided by time- and feature-based regularities

Dynamic information flows, such as speech and music, are composed of rhythmic structures nested across multiple timescales (*Ding et al., 2016*; *Gross et al., 2013*; *Koelsch et al., 2013*; *Peelle and Davis, 2012*). These hierarchical structures are organized in time based on regularities in stimulus timing, that is, when sensory signals are emitted (time-based), as well as regularities in information content, that is, how physical or semantic features of the sensory inputs change over time (feature-based). Accrued evidence suggests that temporal structures formed by time-based regularities are effective in directing attention and enhance information selection at the expected time points (*Jones et al., 2002*; *Nobre et al., 2007*; *Nobre and van Ede, 2018*). Yet the current study demonstrates the role of feature-based temporal structures in recomposing temporal attention deployment, which optimizes the distribution of attentional resources over two temporally proximate targets in the AB task.

We modified the standard AB paradigm by introducing a contextual stream whose physical property changed periodically to form perceivable, but unattended rhythmic cycles in the background. Although this feature-based temporal structure was task-irrelevant, it modulated the deployment of attentional resources along the AB stream, as indicated by higher T2 detection performance when the two targets were located in different cycles than in the same cycle. More intriguingly, this

modulation effect was observed no matter whether the contextual stream was from the auditory (Experiment 1) or the visual (Experiment 2) modality. These findings provide clear evidence that temporal structures defined by periodic changes of physical features in a dynamic context can automatically reconstruct the temporal distribution of visual attention.

In the current study, the rhythmic cycles in the contextual stream consisted of a set of temporally grouped items, some with abrupt changes in physical features across the cycle boundaries. Could the attentional modulation effect be achieved purely on the basis of transient perceptual boundaries or temporal grouping? Findings from several control experiments do not agree with these assumptions. In Experiment 3, the rhythmic cycles of contextual rhythms were defined by cyclic motion without any abrupt changes at the boundaries. Even in this case, the cyclic motion yielded a significant attentional modulation effect, excluding the possibility that the observed effect was caused simply by perceptual changes of the background. In addition, results from Experiments 1d and 1e further confirm that temporal attention guided by temporal regularities rather than perceptual grouping is key to the reduced AB effect. On the one hand, simple grouping without feature-based temporal regularities had little influence on T2 detection (as in Experiment 1d, the feature-based grouping was irregular). On the other hand, when we disrupted time-based regularities by using stochastic stimulus timing, the attentional modulation effect also vanished, even though the rule of feature-based grouping remained in force (as in Experiment 1e, every four identical tones constituted one group). Jointly, these findings point to a mechanism of temporal attentional guidance independent of transient perceptual cues and simple perceptual grouping.

It is worth noting that the attentional modulation effect did not occur in the absence of regular stimulus timing. In other words, the feature-based regularities should work in tandem with the time-based regularities to reconstruct the dynamics of visual temporal attention, at least under the current experimental settings. This finding is consistent with the emerging view concerning the role of a diversity of temporal structures in guiding adaptive behavior (*Nobre and van Ede, 2018*). It has been suggested by studies using auditory materials, mostly in speech and music perception, that temporal regularities embedded in information content can act along with the time-based anticipation in attentional guidance (*Doelling and Poeppel, 2015*; *Morillon et al., 2016*; *Peelle and Davis, 2012*; *Zion Golumbic et al., 2012*). Our findings extend these studies by establishing a mechanism in visual temporal attention that is guided by regularities in feature-defined structures on top of the anticipation based on stimulation timing.

## The roles of dynamic attentional deployment in reducing attentional blink and boosting awareness

The AB phenomenon represents a bottleneck of conscious awareness pertaining to the temporal resolution of visual attention. It is well known for its robustness that even long repetitive training cannot eliminate the AB effect (*Braun, 1998*). Some studies have demonstrated attenuated AB magnitude, as manifested in increased T2 detectability, by enhancing T2 salience with color-salience training (*Choi et al., 2012*), emotional arousal (*Keil and Ihssen, 2004*), or concurrent sounds (*Olivers and Van der Burg, 2008*). Another line of research has also reported improved T2 performance when explicitly cueing the target-onset-asynchrony (TOA) on a trial-by-trial basis (*Martens and Johnson, 2005*) or manipulating the predictability of target onset (*Tang et al., 2014*; *Visser et al., 2015*). Despite implementing different approaches, all these studies tried to manipulate certain aspects of T2, regarding either its salience or predictability in time. By contrast, in our study, the salience of targets and temporal expectations about T2 onset were comparable across all experimental conditions. The only difference between the within- and between-cycle conditions was the positions of the two targets relative to the feature-defined temporal structure. Under this situation, items in the RSVP stream were no longer encoded in isolation, but treated as a part of a structured information flow that could be organized by periodic changes in the context. In particular, when T1 and T2 were separated in different cycles, the temporal relations between them were reframed, which might at least partially reduce the competition between the targets, thus improving the resolution of visual temporal attention and boosting the conscious access to T2. Instead of emphasizing the role of a given target or a certain time point, our findings highlight the significance of attentional deployment as a dynamic process in regulating visual awareness and the AB effect, which is modulated by temporal structures of the entire information flow.

# Neural entrainment to hierarchical contextual rhythms modulates dynamic attending in visual perception

Neural oscillations can be entrained to external rhythms across different frequencies (*Calderone et al., 2014*; *Escoffier et al., 2015*; *Henry et al., 2014*; *Mathewson et al., 2012*; *Schroeder et al., 2010*; *Schroeder and Lakatos, 2009*; *Thut et al., 2011*), allowing the brain to encode dynamic information with multiplexed rhythmic structures across different timescales (*Fontolan et al., 2014*; *Lakatos et al., 2005*; *O'Connell et al., 2015*). A fine example of this comes from studies of speech processing. The linguistic structure possesses a temporal hierarchy—from smaller phonetic elements to larger syllabic and phrasal units, which correspondingly elicit neural entrainment at multiple frequency bands (*Arnal and Giraud, 2012*; *Zion Golumbic et al., 2012*). Moreover, there is growing evidence that cortical tracking of the higher-order temporal structures plays a vital role in speech comprehension (*Ding et al., 2016*; *Gross et al., 2013*; *Peelle and Davis, 2012*). In our EEG study, we demonstrate an analogous entrainment effect that not only keeps track of the original AB stream at 10 Hz but also represents the higher-order feature-based structure of contextual rhythms at 2.5 Hz. This effect, distinct from the hierarchical entrainment to speech signals, does not rely on previously acquired knowledge about the structured information and can be established automatically even when the higher-order structure comes from a task-irrelevant and cross-modal contextual rhythm. More importantly, the magnitude of the 2.5 Hz entrainment effect is significantly correlated with the strength of the attentional modulation effect. The scalp topographic map of correlation is lateralized and restricted to the left parietal region, which was found to be associated with temporal attention (*Bolger et al., 2014*; *Coull and Nobre, 1998*). These findings are in good accordance with our assumption that the cortical tracking of feature-based contextual structure is critical to the redeployment of attentional resources over the AB stream and may lead to the behavioral modulation effect, which sheds fresh light on the adaptive value of the structure-based entrainment effect by expanding its role from rhythmic information (e.g. speech) perception to temporal attention deployment.

There has been a debate about whether the neural alignment to rhythmic stimulation reflects active entrainment of endogenous oscillatory processes (i.e. induced activity) or a series of passively evoked steady-state responses (*Keitel et al., 2019*; *Notbohm et al., 2016*; *Zoefel et al., 2018*). The latter process is also referred to as 'entrainment in a broad sense' by *Obleser and Kayser, 2019*. Given that a presented rhythm always evokes event-related potentials, a better question might be whether the observed alignment reflects the entrainment of endogenous oscillations in addition to evoked steady-state responses. Here, we attempted to tackle this issue by measuring the induced power, which emphasizes the intrinsic non-phase-locked activity, in addition to the phase-locked evoked power. Specifically, we quantified these two kinds of neural activities with the average of single-trial EEG power spectra and the power spectra of trial-averaged EEG signals, respectively, according to *Keitel et al., 2019*. In addition to the observation of evoked responses to the contextual structure, we also demonstrated an attention-related neural tracking of the higher-order temporal structure based on the induced power at 2.5 Hz (see *Figure 4—figure supplement 1*), suggesting that the observed attentional modulation effect is at least partially derived from the entrainment of intrinsic oscillatory brain activity.

In our experiment, the 10 Hz alpha power around T2 is stronger in the between-cycle condition than in the within-cycle condition. A widely accepted function of alpha activity in attention is that alpha oscillations suppress irrelevant visual information during spatial selection (*Kelly et al., 2006*; *Thut et al., 2006*; *Worden et al., 2000*). However, it becomes a controversial issue when there exists rhythmic sensory stimulation at alpha-band, just like the situation in the current study where both the visual stream and the contextual auditory rhythm were emitted at 10 Hz. In such a case, alpha-band neural responses at the stimulation frequency can be interpreted as either passively evoked steady-state responses (SSR) or actively synchronized intrinsic brain rhythms. From the former perspective (i.e. the SSR view), an increase in the amplitude or power at the stimulus frequency may indicate an enhanced attentional allocation to the stimulus stream that may result in better target detection (*Janson et al., 2014*; *Keil et al., 2006*; *Müller and Hübner, 2002*). Conversely, the latter view of the inhibitory function of intrinsic alpha oscillations would produce the opposite prediction. In a previous AB study, *Janson et al., 2014* investigated this issue by separating the stimulus-evoked activity at 12 Hz (using the same power analysis method as ours) from the endogenous

alpha oscillations ranging from 10.35 to 11.25 Hz (as indexed by individual alpha frequency, IAF). Interestingly, they found a dissociation between these two alpha-band neural responses, showing that the RSVP frequency power was higher in non-AB trials (T2 detected) than in AB trials (T2 undetected) while the IAF power exhibited the opposite pattern. According to these findings, the currently observed increase in alpha power for the between-cycle condition may reflect more of the stimulus-driven processes related to attentional enhancement. However, we do not negate the effect of intrinsic alpha oscillations in our study, as the current design is not sufficient to distinguish between these two processes.

Further analysis reveals that, in the left parieto-occipital cluster that exhibits phase-locked neural responses to both feature-based contextual structures at 2.5 Hz and first-order stimulus frequency at 10 Hz, there is an enhancement of phase-amplitude coupling between the delta and alpha oscillations for the between- relative to the within-cycle condition. Moreover, the strength of this delta-alpha coupling enhancement predicts the effect of higher-order temporal structures on dynamic attentional allocation at the individual level. These findings corroborate the idea that neural entrainment to a slower external rhythm may serve as a mechanism of attentional deployment, with the phase of delta oscillation regulating the excitability of neural activity in the alpha band (*Gomez-Ramirez et al., 2011*; *Wilson and Foxe, 2020*; *Wöstmann et al., 2016*).

Taken together, findings from the current study have cast new light on the classic theory of DAT and its neural implementation. The DAT assumes attention to be inherently oscillatory and can be driven by the timing pattern of external events (*Jones, 1976*; *Jones et al., 1982*; *Jones and Boltz, 1989*; *Large and Jones, 1999*). By taking advantage of temporal regularities of isochronous or rhythmic events, attentional synchrony can be established and thus improve perceptual accuracy and elevate response speed. Our study extends the DAT to more general cases of dynamic information processing at both the behavioral and the neural levels. Primarily, our behavioral observations suggest that to utilize regularities in a hierarchical temporal structure, the internal attentional oscillation may not only align with first-order rhythmic structures based on stimulus timing, but also with higher-order rhythmic structures defined by content-based changes of the information flow. Such a dynamic attending process necessitates the synergy between time- and content-based regularities, which could be implemented by neural entrainment to the higher-order temporal structure and its coordination with the cortical tracking of the stimulus rhythm through cross-frequency coupling.

## Conclusion

In summary, the current study emphasizes the role of feature-defined contextual rhythms in reconstructing the deployment of visual attention along dynamic information streams. This work enriches our knowledge, as raised at the beginning of this article, about how we optimize the limited mental capacity to process successive inputs from this ever-changing world. Taking the AB phenomenon as an example, we provide a new perspective on visual temporal attention research—when examining the perception of complex dynamic information, temporal context on multiple timescales should be taken into consideration because it provides a meaningful hierarchical temporal frame for attentional deployment. This temporal frame, implemented by neural entrainment, may serve to organize attentional resources in a prospective manner and help construct our conscious experience of the world in the dimension of time.

## Materials and methods

### Participants

A total of 144 volunteers (aged from 18 to 30 years, 69 females) were recruited and paid for their participation in the current study. One hundred and twenty-eight participated in the behavioral Experiments 1a-1e, 2a-2b, and 3 (16 for each experiment, with participants' gender balanced), and 16 (5 females) in the EEG experiment. All participants had normal or corrected-to-normal vision and normal hearing and were naïve to the purpose of the experiment. Considering the individual differences in the AB effect, only participants who exhibited a typical AB effect (i.e. an impairment of T2 accuracy at short lags compared with that at long lags) during a pre-screening session were asked to take part in the formal experiments. All participants provided written informed consent in

accordance with experimental procedures and protocols approved by the Institutional Review Board of the Institute of Psychology, Chinese Academy of Sciences (ethical approval number: H17028).

## Stimuli

The rapid visual serial presentation (RSVP) stream used in the AB task consisted of 16 items (except in Experiments 1c and 1d). Among these items, one or two were the targets (capital letters selected from the alphabet, excluding B, D, O, I, M, Q, S, W, and Z), and the remaining were distractors (one-digit numbers, 1 and 0 excluded, without repetitions between any two of four successive digits). The items were displayed for 83 ms each and were separated by 17 ms blank intervals (except Experiment 1e), generating a 10 Hz rhythm based on stimulus presentation (see *Figure 1A*, top). Each item subtended 0.47°×0.57° of visual angle and was displayed in white within a gray square (3° × 3°) located at the center of a black screen. In each experiment, a contextual stream, which contained the same number of items as the AB stream but was organized by a feature-defined structure, was presented in synchronization with the AB stream. Stimuli were generated and displayed using MAT-LAB (The MathWorks Inc, Natick, MA) with the Psychophysics toolbox extension (*Brainard, 1997*). Visual stimuli were presented on a 21-inch CRT monitor with a viewing distance of 55 cm in a dim room. Auditory stimuli were delivered binaurally over Bose QC3 headphones with the volume set to a comfortable listening level.

## Procedures

### Behavioral experiments

In all experiments, participants were explicitly instructed to ignore the contextual events and focused attention on the AB task. Participants initiated each trial by pressing the enter key. A white fixation cross appeared for 600 ms at the center of the screen, followed by the presentation of an AB stream (along with an auditory/visual stream in the context session). After the last item disappeared, the central fixation turned blue to remind the participant to report the identities of the target(s) in the order they detected them by typing on the keyboard.

Experiment 1a had a baseline session followed by a context session. In the baseline session, participants viewed only the AB stream and performed the typical AB task. To induce the AB effect, the second target (T2) in the AB stream was located at the second lag of the first target (T1) with a short stimulus onset asynchrony (SOA) of 200 ms, as the magnitude of AB effect is most robust around the second and the third lags. In contrast with the short-SOA condition, we introduced a long-SOA condition where T2 always appeared at the 8th lag of T1 and could rarely be missed. To measure the false alarm rate, we also included catch trials in which only one target was displayed. The context session had the same settings and task as the baseline session, except that a task-irrelevant auditory stream was presented in synchronization with the original RSVP stream. Specifically, the auditory stream was composed of 16 tones, each aligned with the onset of a visual item and displayed for 30 ms. The tone sequence changed its pitch from high (2000 Hz) to low (1200 Hz) or vice versa every four items (corresponding to 400 ms), generating four auditory cycles (i.e. 4-4-4-4) at a rate of 2.5 Hz (*Figure 1A*, middle). To examine the regulation effect of such pitch-defined rhythmic structures, we created two experimental conditions specifically for the short-SOA trials, by varying the positions of T1 and T2 relative to the contextual cycles. In the 'between-cycle condition', T1 and T2 were located in two adjacent cycles; and in the 'within-cycle condition', the two targets were located in the same cycle. To reduce observers' anticipation about the timing of T1 onset across trials, we introduced various T1 positions while keeping T2 located within the middle two cycles. Each session had 120 experimental trials (40 trials for the between-cycle, within-cycle, and long SOA condition each) and 20 catch trials. These trials were divided into four equal blocks, with randomized trial order within each block.

Experiment 1b-1e adopted the same procedure as Experiment 1a but with the following exceptions. In Experiment 1b, as shown in *Figure 1A* (bottom), we abolished the feature-based structure of the contextual streams by pseudo-randomizing the auditory tone sequences while keeping the pitch of tones at target locations the same as that in Experiment 1a. In Experiment 1 c, we changed the temporal structure of the contextual streams by altering their pitch change rate, generating two types of auditory sequences: one with four five-tone cycles displayed at 2 Hz (i.e., 5-5-5-5, see *Figure 2A*, upper), and the other with five three-tone cycles at 3.3 Hz (i.e. 3-3-3-3-3, see *Figure 2A*,

lower). For both frequency conditions, T2 was located in the next to last or third from last cycles. In Experiment 1d, we varied the length of chunks in the contextual streams, generating auditory sequences with four chunks of different lengths (e.g. 5-2-4-3) but always having four tones in the third cycle where the second target appeared (see *Figure 2C*, upper). In Experiment 1e, the feature-based structure remained while the rhythm from stimulus timing was removed (see *Figure 2C*, lower). Specifically, the tone pitch changed every four items just as in Experiment 1a, whereas the stimulus onset asynchrony (SOA) of each visual item was selected randomly from a predetermined uniform distribution (50, 67, 83, 100, 100, 117, 133, 150 ms) to keep the total presentation time identical to that in Experiment 1a. In both Experiment 1d and 1e, T2 was always the second item in the 3rd cycle for the between-cycle condition and the last item in the 3rd cycle for the within-cycle condition.

Experiments 2a and 2b had a design similar to that of Experiments 1a and 1b, except that we replaced the auditory context with a visually presented contextual stream that possessed color-defined temporal structure. Specifically, in the context session of Experiment 2a, the color of the background square changed from green to red or vice versa at the same tempo as that for contextual tones in Experiment 1a (*Figure 3A*, upper). And in Experiment 2b, the background color changed in arrhythmic patterns (*Figure 3C*, upper). Luminance of the two colors was matched for each observer with a chromatic flicker fusion procedure before the experiments.

Experiment 3 consisted of an experimental session with a structured context as that in Experiment 2a and a control session with a random context as that in Experiment 2b. In the experimental session, the contextual rhythm was created by cyclic motion patterns in the background (*Figure 3E*, upper). Specifically, a blue right-angle (width = 0.38°, side length = 1.5°), initiating from one corner (the upper-left or the upper-right, balanced between blocks) of the background square, rotated clockwise at the same pace as the AB stream. In this way, one cycle of rotation corresponded to the appearance of four items (i.e. 400 ms), forming a 2.5 Hz structure based on the motion cycles. In the control session, no cyclic motion pattern remained but the right-angle shifted to a random quadrant under the constraint of identical initial quadrant in each 'cycle' (*Figure 3E*, lower).

Note that in all these experiments, we also labeled the conditions in baseline and control sessions as 'within-cycle' or 'between-cycle', just to indicate that these conditions shared the same absolute target positions with the corresponding conditions in the context session. This design was adopted to control for any potential influence of the absolute position of a target within the AB stream. Specifically, for each experimental condition (within- or between-cycle), we matched the absolute positions of T1 and T2 between the context session and the baseline session without a context (Experiments 1–2), or between the experimental session and the control session with a random context (Experiments 1 and 3).

## EEG experiment

The procedure of the EEG experiment was mostly identical to that of Experiment 1a except for the following modifications. Black items were presented on a gray background and the item size was 0.59°×0.78°. In each trial, the fixation duration was 1000 ms and each item was displayed for 100 ms with no blank interval. T2 was always located within the third cycle of the contextual rhythm. After response, there was a 1.2–1.5 s blank interval. Each subject completed three baseline blocks followed by six experimental blocks with the auditory context. Each block consisted of 40 trials, with 17 short-SOA trials in each of the between- and within-cycle condition, and the remaining six as the catch trials, run in a random order.

## EEG recording

A SynAmps[2] Neuroscan amplifier system (Compumedics Ltd, Abbotsford, Australia) was used for data acquisition. EEG signals were recorded continuously from 64 Ag/AgCl electrodes mounted on an elastic cap according to the extended 10–20 system, with a reference electrode placed between Cz and CPz. Vertical and horizontal eye movements were monitored with two bipolar EOG electrode pairs positioned above and below the left eye and on the outer canthus of each eye. Data were acquired at a sampling rate of 1000 Hz with an online 0.05–100 Hz band-pass filter (notched at 50 Hz). Electrode impedances were kept below 8 kΩ for all electrodes.

## EEG data analysis

### Preprocessing

Data preprocessing and analysis was performed using EEGLAB toolbox (*Delorme and Makeig, 2004*) and FieldTrip (*Oostenveld et al., 2011*) in the MATLAB environment. EEG recordings were down-sampled offline to 500 Hz, high-pass filtered at 0.3 Hz, and then segmented into 2200 ms trials from −600 to 1600 ms relative to the onset of the AB stream. Ocular artifacts were then identified and removed using the ADJUST algorithm (*Mognon et al., 2011*) based on independent component analysis (ICA). Segments with voltage deflections greater than 75 uV were rejected. Residual artifacts were checked by visual inspection. On average, 90 trials remained for each condition and each individual. The segmented data were re-referenced to the average potential of all electrodes excluding the mastoid and EOG electrodes.

### Power analysis

The preprocessed EEG signals were first demeaned by subtracting the average activity of the entire stream over time (i.e. from 0 to 1600 ms) for each epoch, and then averaged across trials for each condition, each participant, and each electrode. Then signals from stream onset were zero-padded and fast Fourier transformed, yielding amplitude and phase estimation at a frequency resolution of 0.5 Hz. Power spectra was calculated as the squared amplitude and then converted to decibel scale (i.e. $10*\log_{10}$). To remove unrelated background noises from the frequency response of stimulus rhythms, for each frequency, the mean power at two nearest neighboring frequencies was subtracted from the power at that center frequency. The subtracted power at each frequency was then averaged across all channels (excluding M1, M2, VEO, HEO, CB1, and CB2) and compared with zero using one-sample $t$ test to determine whether neural oscillations were entrained to temporal structures of the stimulus rhythms. Multiple comparisons across frequencies were controlled by the false discovery rate (FDR, $p < 0.05$) procedure.

### Phase locking analysis

Inter-trial phase coherence (ITPC) serves to indicate the consistency with which intrinsic neural oscillations were phase-locked to the external rhythms over trials. We first obtained phase estimation from spectral decomposition for each single trial based on fast Fourier transform, and then calculated ITPC as follows:

$$\text{ITPC}(f) = \left| \frac{1}{n} \sum_{k=1}^{n} \left( \frac{F_k(f)}{|F_k(f)|} \right) \right| \tag{1}$$

where, for $n$ trials, $F_k(f)$ is the spectral estimate of trial k at frequency $f$, and $\|$ represents the complex norm.

### Time-frequency analysis

In order to measure the neural activity time-locked to T2 at 10 Hz, time-frequency analysis was performed by convolving single-trial data with a complex Morlet tapered wavelet using the *newtimef* function of EEGLAB. To optimize the trade-off between temporal and frequency resolution, the length of wavelets increased linearly from one cycle at the lowest frequency (2 Hz) to 7.5 cycles at the highest frequency (30 Hz, in increments of 0.5 Hz), resulting in power estimates from −321 to 1321 ms around stream onset. For each frequency, power at each time point was then averaged across trials and divided by the average activity in the baseline period from −300 to −100 ms prior to stream onset and log-transformed to decibels. We averaged the alpha powers from 9.5 to 10.5 Hz around the stimulation frequency within the post-T2 (0–100 ms relative to T2 onset) and pre-T2 (-100–0 ms relative to T2 onset) time windows, respectively.

### Delta-alpha phase-amplitude coupling analysis

The modulation index (MI) of phase-amplitude coupling (PAC) was used to measure the coordinative modulation between the phase of ongoing oscillations in delta band (1.5–3.5 Hz) and the power in alpha bands (7–13 Hz) at each electrode. First, the low-frequency phase at delta band ($f_p$) and high-frequency amplitude at alpha band ($f_a$) were estimated by filtering each epoch with a Butterworth

bandpass filter and then applying the Hilbert transform. The broad bandwidth of alpha band (7–13 Hz) was determined to be wide enough to contain the side-bands of the modulating frequency at $f_p$ (2.5 Hz) (*Dvorak and Fenton, 2014*; *Seymour et al., 2017*). Next, the modulation index of PAC was quantified using the mean-vector length method first introduced by *Canolty et al., 2006*. As shown in formula (2), for each epoch, the MI values were calculated by combining low-frequency phase and high-frequency amplitude into complex time series and then taking the length of the average vector within the selected time window (400–1200 ms relative to stream onset), which corresponded to the middle two cycles of the contextual stream. The first and last 400 ms of the stream was discarded to avoid the edge artifacts after bandpass filtering. The resulting MI values were then averaged across trials for each condition.

$$\text{MI} = \left| \frac{1}{N} \sum_{n=1}^{N} A_H(n) e^{i(\Phi_L(n))} \right| \tag{2}$$

where MI is estimated for a single trial with length of N samples or time points, $A_H(n)$ is the amplitude of higher-frequency at time point $n$, $\Phi_L(n)$ is the phase of lower-frequency at time point $n$, and $||$ represents the complex norm.

## Correlation analysis

To examine whether the above EEG indices were associated with the observed attentional modulation effect, we correlated these EEG indices with individual's behavioral modulation index (BMI), which was determined by the following formula:

$$\text{BMI} = \frac{P_{BET} - P_{WIT}}{P_{BET} + P_{WIT}} \tag{3}$$

where $P_{BET}$ and $P_{WIT}$ were the accuracy rate of T2 identification in the between-cycle and the within-cycle conditions in the context session, respectively.

## Cluster-based permutation test

To identify clusters of channels that are significant in each statistical test, we used the cluster-based permutation test, which was first stated by *Maris and Oostenveld, 2007* and used in a number of previous studies (*Doelling and Poeppel, 2015*; *Spaak et al., 2014*). Firstly, cluster-level statistics are calculated as the sum of channel-specific test statistics within every cluster. Then, the maximum of the cluster-level statistics is taken as the actual test statistic. Finally, the significance probability of the maximum cluster-level statistic is evaluated under the permutation distribution obtained with the Monte Carlo method in which the permutation cluster-level statistic is calculated by randomly swapping the conditions in participants 1000 times.

# Acknowledgements

This research was supported by grants from the National Natural Science Foundation of China (31830037 and 31771211), the Strategic Priority Research Program (XDB32010300) and the Youth Innovation Promotion Association (2018116) of the Chinese Academy of Sciences, the National Key Research and Development Project (2020AAA0105600), and the Fundamental Research Funds for the Central Universities.

# Additional information

### Funding

| Funder | Grant reference number | Author |
| --- | --- | --- |
| National Natural Science Foundation of China | 31830037 | Yi Jiang |
| National Natural Science Foundation of China | 31771211 | Ying Wang |

| | | |
|---|---|---|
| Chinese Academy of Sciences | XDB32010300 | Yi Jiang |
| Chinese Academy of Sciences | 2018116 | Ying Wang |
| Ministry of Science and Technology of the People's Republic of China | 2020AAA0105600 | Yi Jiang |

The funders had no role in study design, data collection and interpretation, or the decision to submit the work for publication.

## Author contributions

Peijun Yuan, Data curation, Software, Formal analysis, Validation, Investigation, Visualization, Methodology, Writing - original draft; Ruichen Hu, Software, Writing - original draft; Xue Zhang, Software; Ying Wang, Conceptualization, Supervision, Funding acquisition, Visualization, Methodology, Writing - review and editing; Yi Jiang, Supervision, Funding acquisition, Methodology, Writing - review and editing

## Author ORCIDs

Ying Wang (iD) https://orcid.org/0000-0002-5756-2480
Yi Jiang (iD) https://orcid.org/0000-0002-5746-7301

## Ethics

Human subjects: All participants provided written informed consent in accordance with experimental procedures and protocols approved by the Institutional Review Board of the Institute of Psychology, Chinese Academy of Sciences (ethical approval number: H17028).

## Decision letter and Author response

Decision letter https://doi.org/10.7554/eLife.65118.sa1
Author response https://doi.org/10.7554/eLife.65118.sa2

## Additional files

### Supplementary files

• Transparent reporting form

### Data availability

We have provided the behavioral and EEG data for individual participants as additional data files. Source data files for Figures 1-5 have been uploaded to the Open Science Framework (https://osf.io/4xzv7/).

The following dataset was generated:

| Author(s) | Year | Dataset title | Dataset URL | Database and Identifier |
|---|---|---|---|---|
| Peijun Y | 2021 | Cortical entrainment to hierarchical contextual rhythms recomposes dynamic attending in visual perception | https://osf.io/4xzv7/ | Open Science Framework, 4xzv7 |

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
