## [Decision Letter]

**Acceptance summary:**

This study, by using a series of behavioral experiments and EEG recordings, demonstrates that attention is essentially modulated by higher-order rhythmic regularity, supporting the notion that rhythmic context implements a second-order temporal structure to the first-order regularities posited in dynamic attention theory.

**Decision letter after peer review:**

Thank you for submitting your article "Cortical entrainment to hierarchical contextual rhythms recomposes dynamic attending in visual perception" for consideration by *eLife*. Your article has been reviewed by 3 peer reviewers, one of whom is a member of our Board of Reviewing Editors, and the evaluation has been overseen by Floris de Lange as the Senior Editor. The reviewers have opted to remain anonymous.

All the reviewers agree that it is a well-designed study and the results are exciting by using an innovative approach to modulate attention blink. The reviewers have raised several major concerns that need the authors to address and do additional analysis.

Evaluation Summary: This study by Wang et al. used a series of carefully designed behavioral experiments to convincingly demonstrate that the attentional blink (AB) could be modulated by higher-order rhythmic regularity. EEG results further support the link between the elicited neural entrainment and the AB modulation effect. They propose that the rhythmic context implements a second-order temporal structure to the first-order regularities posited in dynamic attention theory.

Essential Revisions:

1. The current AB behavior results lack several key aspects shown in typical AB experiments. First, typical AB effect would not just test one lag as did here. The authors should show the behavioral results for several lags, i.e., a full AB curve. The results would be informative as to how cortical entrainment affects the whole AB curve. Moreover, there is no data regarding T1 performance, and it is important to show that the better performance for T2 is not due to worse performance in detecting T1. Finally, typical AB design would examine T2 performance when T1 is ignored relative to when T1 has to be detected, but the data is not provided here.

2. About the EEG results. A general concern is whether the observed behavioral related neural index, e.g., alpha-band power, cross-frequency coupling, could be simply explained in terms of ERP response for T2. For example, when the ERP response for T2 is larger for between-chunk condition compared to within-chunk condition, the alpha-power for T2 would be also larger for between-chunk condition. Likewise, this might also explain the cross-frequency coupling results. The authors should do more control analyses to address the possibility, e.g., plotting the ERP response for the two conditions and regressing them out from the oscillatory index.

3. The alpha-band increase for T2 is indeed contradictory to the well-known inhibitory function of alpha-band in attention. How could a target that is better discriminated elicit stronger inhibitory response? Related to the above point, the observed enhancement in alpha-band power and its coupling to low-frequency oscillation might derive from an enhanced ERP response for T2 target.

4. To support that it is the context-induced entrainment that leads to the modulation in AB effect, the authors could examine pre-T2 response, e.g., alpha-power, and cross-frequency coupling, as well as its relationship to behavioral performance. The pre-stimulus response might be more convincing to support the authors' claim.

5. About the entrainment to rhythmic context and its relation to behavioral modulation index. Previous studies have demonstrated the hierarchical temporal structure in speech signals, e.g., emergence of word-level entrainment introduced by language experience. Therefore, it is well expected that imposing a second-order structure on a visual stream would elicit the corresponding steady-state response. The authors should add more discussions explaining how their findings contribute new understandings to the neural mechanism for the intriguing phenomena.

---

## [Author Response]

Essential Revisions:1. The current AB behavior results lack several key aspects shown in typical AB experiments. First, typical AB effect would not just test one lag as did here. The authors should show the behavioral results for several lags, i.e., a full AB curve. The results would be informative as to how cortical entrainment affects the whole AB curve. Moreover, there is no data regarding T1 performance, and it is important to show that the better performance for T2 is not due to worse performance in detecting T1. Finally, typical AB design would examine T2 performance when T1 is ignored relative to when T1 has to be detected, but the data is not provided here.

We appreciate the reviewer for his/her thoughtful comments. To demonstrate the AB effect, we did include two T2 lag conditions in our study (Experiments 1a, 1b, 2a, and 2b) – a short-SOA condition where T2 was located at the second lag of T1 (i.e., SOA = 200 ms), and a long-SOA condition where T2 appeared at the 8th lag of T1 (i.e., SOA = 800 ms). In a typical AB effect, T2 performance at short lags is remarkably impaired compared with that at long lags. In our study, we consistently replicated this effect across the experiments, as reported in the Results section of Experiment 1 (page 5, line 106). Overall, the T2 detection accuracy conditioned on correct T1 response was significantly impaired in the short-SOA condition relative to that in the long-SOA condition (mean accuracy > 0.9 for all experiments), during both the context session and the baseline session. More crucially, when looking into the magnitude of the AB effect as measured by (ACC_long-SOA_ – ACC_short-SOA_)/ACC_long-SOA_, we still obtained a significant attentional modulation effect (for Experiment 1a, *t*(15) = -2.729, *p* = .016, Cohen’s *d* = 0.682; for Experiment 2a, *t*(15) = -4.143, *p* <.001, Cohen’s *d* = 1.036) similar to that reflected by the short-SOA condition alone, further confirming that cortical entrainment effectively influences the AB effect.

Although we included both the long- and short-SOA conditions in the current study, we focused on T2 performance in the short-SOA condition rather than along the whole AB curve for the following reasons. Firstly, for the long-SOA conditions, the T2 performance is at ceiling level, making it an inappropriate baseline to probe the attentional modulation effect. We focused on Lag 2 because previous research has identified a robust AB effect around the second lag (Raymond et al., 1992), which provides a reasonable and sensitive baseline to probe the potential modulation effect of the contextual auditory and visual rhythms. Note that instead of using multiple lags, we varied the length of the rhythmic cycles (i.e., a cycle of 300 ms, 400 ms, and 500 ms corresponding to a rhythm frequency of 3.3 Hz, 2.5 Hz, and 2 Hz, respectively, all within the delta band), and showed that the attentional modulation effect could be generalized to these different delta-band rhythmic contexts, regardless of the absolute positions of the targets within the rhythmic cycles.

As to the T1 performance, the overall accuracy was very high, ranging from 0.907 to 0.972, in all of our experiments. The corresponding results have been added to the Results section of the revised manuscript (page 5, line 103). Notably, we did not find T1-T2 trade-offs in most of our experiments, except in Experiment 2a where T1 performance showed a moderate decrease in the between-cycle condition relative to that in the within-cycle condition (mean ± SE: 0.888 ± 0.026 vs. 0.933 ± 0.016, respectively; *t*(15) = -2.217, *p* = .043). However, by examining the relationship between the modulation effects (i.e., the difference between the two experimental conditions) on T1 and T2, we did not find any significant correlation (*p* = .403), suggesting that the better performance for T2 was not simply due to the worse performance in detecting T1.

Finally, previous studies have shown that ignoring T1 would lead to ceiling-level T2 performance (Raymond et al., 1992). Therefore, we did not include such manipulation in the current study, as in that case, it would be almost impossible for us to detect any contextual modulation effect.

2. About the EEG results. A general concern is whether the observed behavioral related neural index, e.g., alpha-band power, cross-frequency coupling, could be simply explained in terms of ERP response for T2. For example, when the ERP response for T2 is larger for between-chunk condition compared to within-chunk condition, the alpha-power for T2 would be also larger for between-chunk condition. Likewise, this might also explain the cross-frequency coupling results. The authors should do more control analyses to address the possibility, e.g., plotting the ERP response for the two conditions and regressing them out from the oscillatory index.

Thanks for the comments. In general, the rhythmic stimulation in the AB paradigm prevents EEG signals from returning to the baseline. Therefore, we cannot observe typical ERP components purely related to individual items, except for the P1 and N1 components related to the stream onset, which reveals no difference between the two conditions and are trailed by steady-state responses (SSRs) resonating at the stimulus rate (Author response image 1).

To further inspect the potential differences in the target-related ERP signals between the within- and between-cycle conditions, we plotted the target-aligned waveforms for these experimental conditions. As shown in Author response image 2, a drop of ERP amplitude occurred for both conditions around T2 onset, and the difference between these two conditions was not significant (paired *t*-test estimated on mean amplitude every 20 ms from 0 to 700 ms relative to T1 onset, *p* >.05, FDR-corrected).

Since there is a trend of enhanced ERP response for the between-cycle relative to the within-cycle condition during the period of 0 to 100 ms after T2 onset (paired *t*-test on mean amplitude, *p* = .065, uncorrected), we then directly examined whether such post-T2 responses contribute to the behavioral attentional modulation effect and behavior-related neural indices. Crucially, we did not find any significant correlation of such T2-related ERP enhancement with the behavioral modulation index (BMI), or with the reported effects of alpha power and cross-frequency coupling (PAC). Furthermore, after controlling for the T2-related ERP responses, there still remains a significant correlation between the delta-alpha PAC and the BMI (*r*_partial_ = .596, *p* = .019), which is not surprising given that the PAC is calculated based on an 800-ms time window covering more pre-T2 than post-T2 periods (see the response to point #4 for details) rather than around the T2 onset. Taken together, these results clearly suggest that the T2-related ERP responses cannot explain the attentional modulation effect and the observed behavior-related neural indices.

**Author response image 1. sa2fig1:** ERPs aligned to stream onset. EEG signals were filtered between 1–30 Hz, baseline-corrected (-200 to 0 ms before stream onset), and averaged across the electrodes in left parieto-occipital area where 10-Hz alpha power showed attentional modulation effect.

**Author response image 2. sa2fig2:** ERPs aligned to T1 onset. EEG signals were filtered between 1–30 Hz, and baseline-corrected using signals -100 to 0 ms before T1 onset. The two dash lines indicate the onset of T1 and T2, respectively.

3. The alpha-band increase for T2 is indeed contradictory to the well-known inhibitory function of alpha-band in attention. How could a target that is better discriminated elicit stronger inhibitory response? Related to the above point, the observed enhancement in alpha-band power and its coupling to low-frequency oscillation might derive from an enhanced ERP response for T2 target.

Thanks for the comment. A widely accepted function of alpha activity in attention is that alpha oscillations suppress irrelevant visual information during spatial selection (Kelly et al., 2006; Thut et al., 2006; Worden et al., 2000). However, it becomes a controversial issue when there exists rhythmic sensory stimulation at alpha-band, just like the situation in the current study where both the visual stream and the contextual auditory rhythm were emitted at 10 Hz. In such a case, alpha-band neural responses at the stimulation frequency can be interpreted as either passively evoked steady-state responses (SSR) or actively synchronized intrinsic brain rhythms. From the former perspective (i.e., the SSR view), an increase in the amplitude or power at the stimulus frequency may indicate an enhanced attentional allocation to the stimulus stream that may result in better target detection (Janson et al., 2014; Keil et al., 2006; Müller and Hübner, 2002). Conversely, the latter view of the inhibitory function of intrinsic alpha oscillations would produce the opposite prediction. In a previous AB study, Janson and colleagues (2014) investigated this issue by separating the stimulus-evoked activity at 12 Hz (using the same power analysis method as ours) from the endogenous alpha oscillations ranging from 10.35 to 11.25 Hz (as indexed by individual alpha frequency, IAF). Interestingly, they found a dissociation between these two alpha-band neural responses, showing that the RSVP frequency power was higher in non-AB trials (T2 detected) than in AB trials (T2 undetected) while the IAF power exhibited the opposite pattern. According to these findings, the currently observed increase in alpha power for the between-cycle condition may reflect more of the stimulus-driven processes related to attentional enhancement. However, we do not negate the effect of intrinsic alpha oscillations in our study, as the current design is not sufficient to distinguish between these two processes. We have discussed this point in the revised manuscript (page 18, line 477). Also, we have to admit that “alpha power” may not be the most precise term to describe our findings of the stimulus-related results. Thus, we have specified it as “neural responses to first-order rhythms at 10 Hz” and “10 Hz alpha power” in the revised manuscript (see page 12 in the Results section and page 18 in the Discussion section).

As for the contribution of T2-related ERP response to the observed effect of 10 Hz power and cross-frequency coupling, please refer to our response to point #2.

4. To support that it is the context-induced entrainment that leads to the modulation in AB effect, the authors could examine pre-T2 response, e.g., alpha-power, and cross-frequency coupling, as well as its relationship to behavioral performance. The pre-stimulus response might be more convincing to support the authors' claim.

Thanks for the insightful suggestion. Following this suggestion, we have examined the 10 Hz alpha power within the time window of -100–0 ms before T2 onset and found stronger activity for the between-cycle condition than for the within-cycle condition. This pre-T2 response is similar to the post-T2 response except that it is more restricted to the left parieto-occipital cluster (CP3, CP5, P3, P5, PO3, PO5, POZ, O1, OZ, *t*(15) = 2.774, *p* = .007), which partially overlaps with the cluster that exhibits a delta-alpha coupling effect significantly correlated with the BMI. We have incorporated these findings into the main text (page 12, line 315) and the Figure 5A of the revised manuscript.

As for the coupling results reported in our manuscript, the coupling index (PAC) was calculated based on the activity during the second and third cycles (i.e., 400 to 1200 ms from stream onset) of the contextual rhythm, most of which covers the pre-T2 period as T2 always appeared in the third cycle for both conditions. Together, these results on pre-T2 10 Hz alpha power and cross-frequency coupling, as well as its relationship to behavioral performance, jointly suggest that the observed modulation effect is caused by the context-induced entrainment rather than being a by-product of post-T2 processing.

5. About the entrainment to rhythmic context and its relation to behavioral modulation index. Previous studies have demonstrated the hierarchical temporal structure in speech signals, e.g., emergence of word-level entrainment introduced by language experience. Therefore, it is well expected that imposing a second-order structure on a visual stream would elicit the corresponding steady-state response. The authors should add more discussions explaining how their findings contribute new understandings to the neural mechanism for the intriguing phenomena.

Thanks for the suggestion. We have provided more discussion on this important issue in the revised manuscript (page 17, line 447). In brief, our study demonstrates how cortical tracking of feature-based hierarchical structure reframes the deployment of attentional resources over visual streams. This effect, distinct from the hierarchical entrainment to speech signals (Ding et al., 2016; Gross et al., 2013), does not rely on previously acquired knowledge about the structured information and can be established automatically even when the higher-order structure comes from a task-irrelevant and cross-modal contextual rhythm. On the other hand, our finding sheds fresh light on the adaptive value of the structure-based entrainment effect by expanding its role from rhythmic information (e.g., speech) perception to temporal attention deployment. To our knowledge, few studies have tackled this issue in visual or speech processing.

References:

Raymond, J. E., Shapiro, K. L., and Arnell, K. M. (1992). Temporary suppression of visual processing in an RSVP task: An attentional blink? Journal of Experimental Psychology: Human Perception and Performance, 18(3), 849–860. https://doi.org/10.1037/0096-1523.18.3.849

Janson, J., De Vos, M., Thorne, J. D., and Kranczioch, C. (2014). Endogenous and Rapid Serial Visual Presentation-induced Alpha Band Oscillations in the Attentional Blink. Journal of Cognitive Neuroscience, 26(7), 1454–1468. https://doi.org/10.1162/jocn_a_00551

Keil, A., Ihssen, N., and Heim, S. (2006). Early cortical facilitation for emotionally arousing targets during the attentional blink. BMC Biology, 4(1), 23. https://doi.org/10.1186/1741-7007-4-23

Kelly, S. P., Lalor, E. C., Reilly, R. B., and Foxe, J. J. (2006). Increases in Alpha Oscillatory Power Reflect an Active Retinotopic Mechanism for Distracter Suppression During Sustained Visuospatial Attention. Journal of Neurophysiology, 95(6), 3844–3851. https://doi.org/10.1152/jn.01234.2005

Müller, M. M., and Hübner, R. (2002). Can the Spotlight of Attention Be Shaped Like a Doughnut? Evidence From Steady-State Visual Evoked Potentials. Psychological Science, 13(2), 119–124. https://doi.org/10.1111/1467-9280.00422

Thut, G., Nietzel, A., Brandt, S., and Pascual-Leone, A. (2006). Alpha-band electroencephalographic activity over occipital cortex indexes visuospatial attention bias and predicts visual target detection. The Journal of Neuroscience : The Official Journal of the Society for Neuroscience, 26(37), 9494–9502. https://doi.org/10.1523/JNEUROSCI.0875-06.2006

Worden, M. S., Foxe, J. J., Wang, N., and Simpson, G. V. (2000). Anticipatory Biasing of Visuospatial Attention Indexed by Retinotopically Specific α-Bank Electroencephalography Increases over Occipital Cortex. Journal of Neuroscience, 20(6), RC63–RC63. https://doi.org/10.1523/JNEUROSCI.20-06-j0002.2000

Ding, N., Melloni, L., Zhang, H., Tian, X., and Poeppel, D. (2016). Cortical tracking of hierarchical linguistic structures in connected speech. Nature Neuroscience, 19(1), 158–164. https://doi.org/10.1038/nn.4186

Gross, J., Hoogenboom, N., Thut, G., Schyns, P., Panzeri, S., Belin, P., and Garrod, S. (2013). Speech Rhythms and Multiplexed Oscillatory Sensory Coding in the Human Brain. PLoS Biol, 11(12). https://doi.org/10.1371/journal.pbio.1001752